

# T cell receptor signaling pathway and cytokine-cytokine receptor interaction affect the rehabilitation process after respiratory syncytial virus infection

Zuanhao Qian[*], Zhenglei Zhang[*] and Yingying Wang

Department of Pediatrics, Taikang Xianlin Drum Tower Hospital, Nanjing, China
[*] These authors contributed equally to this work.

## ABSTRACT

**Background**. Respiratory syncytial virus (RSV) is the main cause of respiratory tract infection, which seriously threatens the health and life of children. This study is conducted to reveal the rehabilitation mechanisms of RSV infection.

**Methods**. E-MTAB-5195 dataset was downloaded from EBI ArrayExpress database, including 39 acute phase samples in the acute phase of infection and 21 samples in the recovery period. Using the limma package, differentially expressed RNAs (DE-RNAs) were analyzed. The significant modules were identified using WGCNA package, and the mRNAs in them were conducted with enrichment analysis using DAVID tool. Afterwards, co-expression network for the RNAs involved in the significant modules was built by Cytoscape software. Additionally, RSV-correlated pathways were searched from Comparative Toxicogenomics Database, and then the pathway network was constructed.

**Results**. There were 2,489 DE-RNAs between the two groups, including 2,386 DE-mRNAs and 103 DE-lncRNAs. The RNAs in the black, salmon, blue, tan and turquoise modules correlated with stage were taken as RNA set1. Meanwhile, the RNAs in brown, blue, magenta and pink modules related to disease severity were defined as RNA set2. In the pathway networks, *CD40LG* and *RASGRP1* co-expressed with *LINC00891/LINC00526/LINC01215* were involved in the T cell receptor signaling pathway, and *IL1B*, *IL1R2*, *IL18*, and *IL18R1* co-expressed with *BAIAP2-AS1/CRNDE/LINC01503/SMIM25* were implicated in cytokine-cytokine receptor interaction.

**Conclusion**. *LINC00891/LINC00526/LINC01215* co-expressed with *CD40LG* and *RASGRP1* might affect the rehabilitation process of RSV infection through the T cell receptor signaling pathway. Besides, *BAIAP2-AS1/CRNDE/LINC01503/SMIM25* co-expressed with *IL1* and *IL18* families might function in the clearance process after RSV infection via cytokine-cytokine receptor interaction.

Corresponding author
Zuanhao Qian,
qianzuanhao@163.com

## INTRODUCTION

Respiratory syncytial virus (RSV) belongs to the paramyxovirus family of pneumonia genus, which is the most common cause of respiratory tract infection in infants (*Hall et al., 2009*). RSV can induce interstitial pneumonia and bronchiolitis, and RSV infection is a disease that seriously endangers the health and life of children (*Fauroux et al., 2017*). At present, the main treatment methods of RSV infection include oxygen therapy, fogging machine antivirus therapy, and antibacterial drug treatment (*Handforth, Sharland & Friedland, 2004*). Approximately 60% of infants will be infected with RSV during their first season of the virus, and almost all children experience RSV infection at the age of 2–3 years old in the United States (*Hama et al., 2015*; *Louis et al., 2016*). Therefore, exploring the possible mechanisms of RSV infection is of great importance.

Through specific antagonistic effects on host protein functions, induction of RNA stress particles and induction of changes in host gene expression patterns, RSV can change the transcription of host genes and the translation of host transcripts (*Tripp, Mejias & Ramilo, 2013*). Long noncoding RNAs (lncRNAs) are a class of RNAs with a length greater than 200 nt and lacking the ability to encode proteins (*Andreia, Marc & George, 2015*). At present, lncRNAs are found to have complex functions and can be involved in all stages of gene expression regulation (*Wahlestedt, 2013*; *Wang et al., 2011*). LncRNA maternally expressed 3 (*MEG3*) expression is decreased in RSV-infected nasopharyngeal samples, and *MEG3* can suppress RSV infection via inhibiting toll-like receptor 4 (TLR4) signaling (*Tao et al., 2018*). Through mediating E2F transcription factor 3 (*E2F3*) expression, lncRNA plasmacytoma variant translocation 1 (*PVT1*) is deemed to be correlated with the effects of $\alpha$-asarone in the treatment of RSV-induced asthma (*Yu et al., 2017*). Additionally, transcription elongation is involved in IFN-stimulated gene (*ISG*) expression induced by RSV, and cyclin-dependent kinase 9 (*CDK9*) activity may serve as a potential target for immunomodulation in RSV-associated lung disease (*Tian et al., 2013*). Nevertheless, the pathogenesis of RSV infection has not been fully understood.

*Jong et al. (2016)* performed a blood transcriptome profiling analysis and identified an 84 gene signature that could predict the course of RSV disease. However, they had not performed in-depth bioinformatics analyses to explore the rehabilitation mechanisms of RSV infection. Using the dataset deposited by *Jong et al. (2016)*, the lncRNA and mRNA expression profiles of RSV-infected blood samples from infants were compared to screen the molecules markers between acute phase and recovery period of RSV infection. This study might contribute to investigating the gene expression changes in blood from acute phase to recovery period of RSV infection and revealing the underlying mechanisms of disease rehabilitation.

## MATERIALS AND METHODS

### Data source and data preprocessing

The E-MTAB-5195 dataset was acquired from EBI ArrayExpress database (https://www.ebi.ac.uk/arrayexpress/) (*Brazma et al., 2003*), which was detected on the platform of Affymetrix GeneChip Human Genome U133 Plus 2.0. E-MTAB-5195 contained 60

blood samples from infants, including 39 samples in the acute phase of infection (14 females and 25 males; mean age = 146.49 days) and 21 samples in the recovery period (11 females and 10 males; mean age = 137.95 days). This study analyzed the expression profile dataset obtained from public database, and no animal or human experiments were involved. Therefore, no ethical review or informed consents were needed.

The raw expression profile data in .CEL format were conducted with format transformation, filling of missing data (median method), background correction (MicroArray Suite method), and data standardization (quantile method) using the R package oligo (version 1.41, http://www.bioconductor.org/packages/release/bioc/html/oligo.html) (*Parrish & Spenceriii, 2004*).

## Differential expression analysis

The platform information of E-MTAB-5195 was downloaded. Combined with the Transcript ID, RefSeq ID, and location information provided in platform information, the mRNAs and lncRNAs in the expression profile E-MTAB-5195 were annotated based on the 19,198 protein coding genes and 3909 lncRNAs included in HUGO Gene Nomenclature Committee (HGNC, http://www.genenames.org/) database (*Bruford et al., 2008*).

According to disease stage information, the samples were divided into samples in the acute phase of infection and samples in the recovery period. Using the R package limma (version 3.34.0, https://bioconductor.org/packages/release/bioc/html/limma.html) (*Ritchie et al., 2015*), differential expression analysis for the two groups were carried out. The false discovery rate (FDR) < 0.05 and |log fold change (FC)| > 0.263 were defined as the criteria for screening differentially expressed RNAs (DE-RNAs). Based on the expression levels of the screened RNAs, the R package pheatmap (version 1.0.8, *Kolde, 2015*) (*Wang et al., 2014*) was used to perform hierarchical clustering analysis (*Szekely & Rizzo, 2005*) and present the result in clustering heatmap.

## Weighed gene co-expression network analysis (WGCNA) and enrichment analysis

The systems biology method WGCNA can be applied for integrating gene expression and identify disease-associated modules (*Li et al., 2018*). Using the R package WGCNA (version 1.61, *Langfelder & Horvath, 2008*) (*Langfelder & Horvath, 2008*), the DE-RNAs were analyzed to identify the modules significantly related to disease states and clinical factors. The analysis procedures included assumption of the scale-free network, definition of co-expression correlation matrix, definition of adjacency function, calculation of dissimilarity coefficients, and identification of disease-associated modules. The modules having significant correlation with disease symptoms (*p*-value < 0.05) and higher correlation coefficients compared with the control (grey module) were taken as the significant modules.

Then, DAVID online tool (version 6.8, https://david.ncifcrf.gov/) (*Lempicki, 2008*) was utilized to perform Gene Ontology (GO) (*The Gene Ontology Consortium, 2015*) and Kyoto Encyclopedia of Genes and Genomes (KEGG) (*Kanehisa et al., 2016*) enrichment analyses for the mRNAs in the significant modules. The *p*-value < 0.05 was selected as the threshold of enrichment significance.

## Co-expression network analysis

For the lncRNAs and mRNAs in the significant modules, pearson correlation coefficients (PCCs) (*Hauke & Kossowski, 2011*) were calculated using the cor function in R (https://www.rdocumentation.org/packages/stats/versions/3.6.0/topics/cor). Afterwards, visualization of co-expression network was conducted using Cytoscape software (http://www.cytoscape.org/) (*Kohl, Wiese & Warscheid, 2011*). Moreover, KEGG pathways were enriched for the mRNAs in the co-expression network using DAVID tool (*Lempicki, 2008*).

## Network analysis for the RSV-correlated pathways

Using "Respiratory Syncytial Virus" as keyword, the KEGG pathways directly related to RSV were searched form Comparative Toxicogenomics Database (2017 update, http://ctd.mdibl.org/) (*Davis et al., 2013*). By comparing the searched pathways and the pathways enriched for the mRNAs in the co-expression network, the overlapped pathways and the mRNAs involved in them were obtained. Furthermore, the clusters (including both lncRNAs and mRNAs) of the RNAs involved in the RSV-correlated pathways were extracted from the co-expression network to construct the network of RSV-correlated pathways.

# RESULTS

## Differential expression analysis

The raw data of expression profile dataset E-MTAB-5195 were preprocessed (Fig. 1). After annotation was performed based on platform information, a total of 16,984 protein coding RNAs and 939 lncRNAs were obtained. Then, the samples were grouped according to disease stage information. There were a total of 2,489 DE-RNAs between the two groups, including 2,386 DE-mRNAs (1,393 up-regulated and 993 down-regulated) and 103 DE-lncRNAs (62 up-regulated and 41 down-regulated) (Fig. 2). The clustering heatmap of the DE-RNAs indicated that the DE-RNAs could separate the two groups of samples and thus had sample characteristics (Fig. 3).

## WGCNA and enrichment analysis

The 2,489 DE-RNAs were further analyzed and screened using WGCNA. In order to satisfy the premise of scale-free network distribution as far as possible, the value of the weight parameter "power" of adjacency matrix needed to be explored. The selection range of network construction parameters was set, and the scale-free topological matrix was calculated. The value of "power" when the square value of the correlation coefficient reached 0.9 for the first time was selected, namely, power = 18. At this time, the average node degree of the constructed co-expression network was 1, which fully conformed to the properties of small world network (Fig. 4A). Then, the dissimilarity coefficients between the gene points were calculated, and the system cluster tree was obtained. The minimum number of genes was set as 70 for each module, and the pruning height was set as cutHeight = 0.99. A total of 13 modules (except of grey module) (Fig. 4B), and the number and type of RNAs involved in each module were listed in Table 1. Finally, the correlation between each

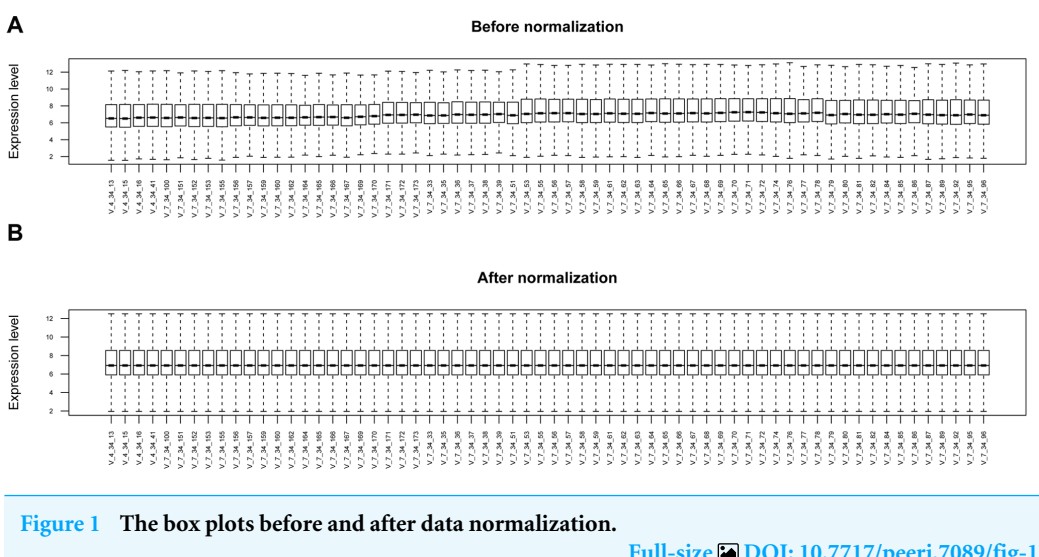

**Figure 1   The box plots before and after data normalization.**

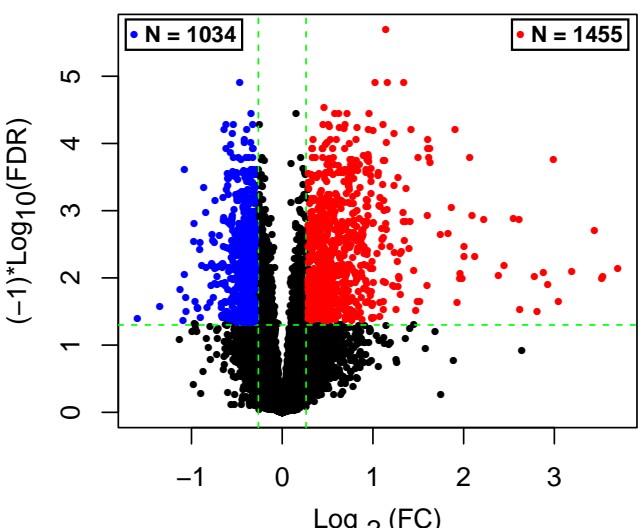

**Figure 2   The volcano plot of the differentially expressed RNAs (DE-RNAs).** The green horizontal dashed line represents false discovery rate (FDR) < 0.05, the two green vertical dashed lines represent—log fold change (FC)—> 0.263. The blue and red dots separately represent the significantly down-regulated and up-regulated RNAs in the samples in the acute phase of infection.

module and different disease symptoms were calculated. The black, salmon, blue, tan and turquoise modules were significantly correlated with stage (acute/recovery), meanwhile, brown, blue, magenta and pink modules were significantly related to disease severity (Mild/Moderate/Severe) (Fig. 4C). The RNAs in black, salmon, blue, tan and turquoise modules were taken as RNA set1 correlated with stage. Similarly, the RNAs in brown, blue, magenta and pink modules were considered as RNA set2 associated with disease severity.

The mRNAs in RNA set1 and RNA set2 separately were conducted with enrichment analysis. For the mRNAs in RNA set1, 17 GO terms (such as regulation of transcription

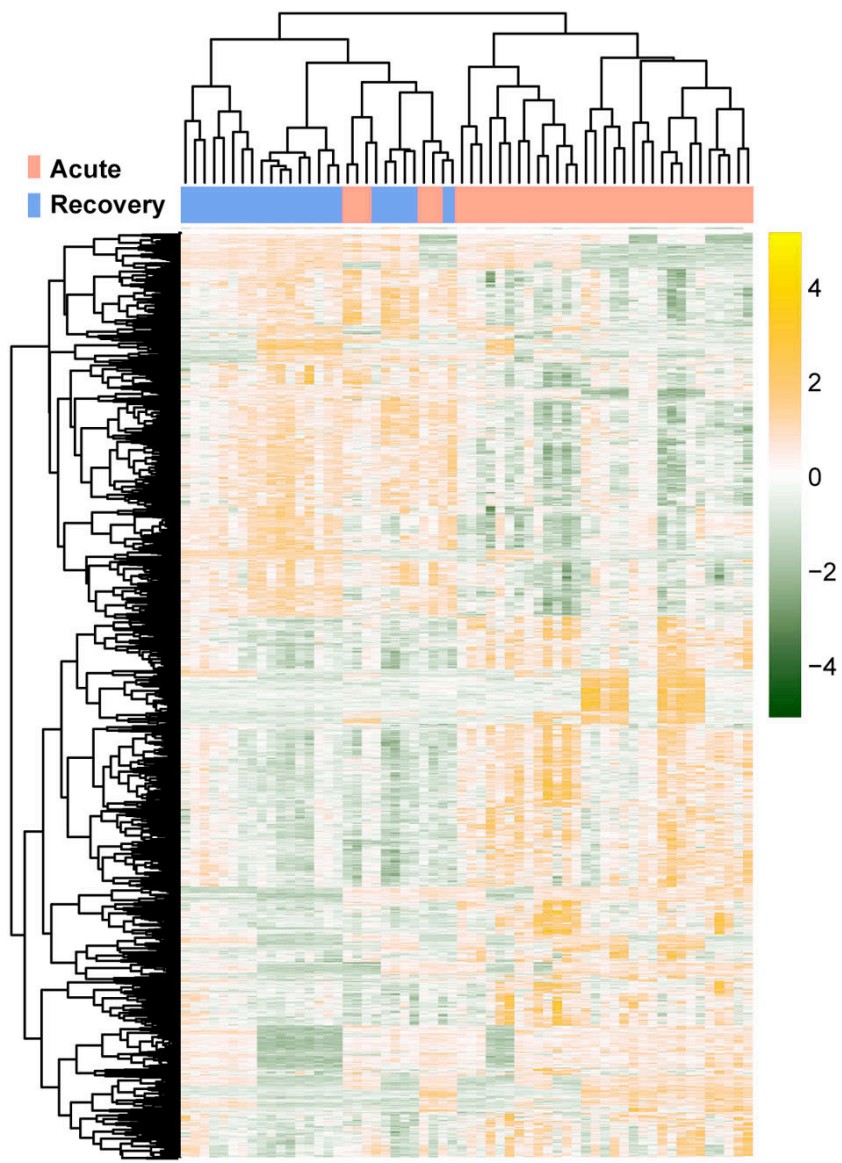

**Figure 3 The lustering heatmap based on the expression levels of the differentially expressed RNAs (DE-RNAs).** Red and blue in the sample bar represent the samples in the acute phase of infection and the samples in the recovery period, respectively.

and lymphocyte activation) and seven KEGG pathways (such as cell adhesion molecules (CAMs) and primary immunodeficiency) were enriched. For the mRNAs in RNA set2, 20 GO terms (including defense response and response to wounding) and eight KEGG pathways (including cytokine-cytokine receptor interaction and lysosome) were predicted (Table 2).

## Co-expression network analysis

For the RNA set1 and RNA set2, the PCCs between the lncRNAs and mRNAs separately were calculated and the lncRNA-mRNA pairs with PCC >0.6 were remained. Subsequently,

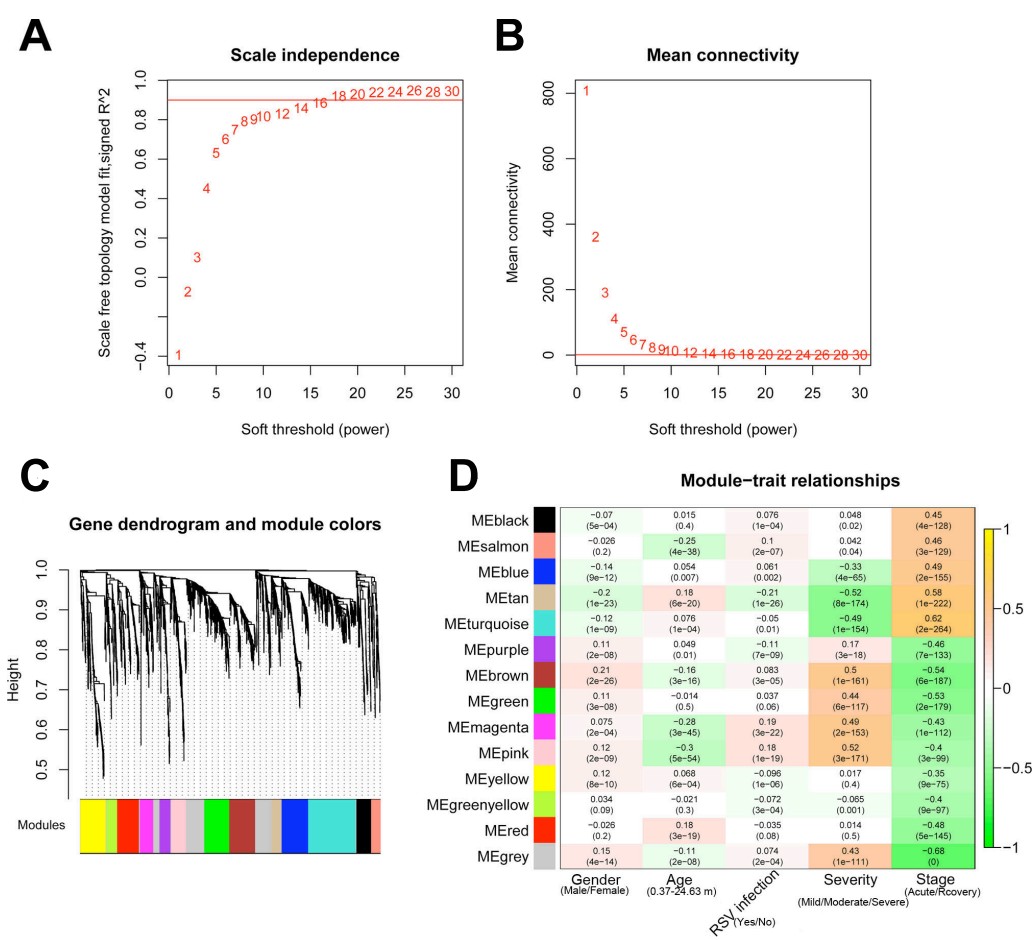

**Figure 4** **The results of weighed gene co-expression network analysis (WGCNA).** (A) The selection of the weight parameter "power" of adjacency matrix (left diagram; red line is the standard line when the square value of the correlation coefficient achieved 0.9) and the mean connectivity of RNAs (right diagram; red line indicates that the average node degree of the constructed co-expression network is 1 when power = 18); (B) The tree diagram for module division (different color represent different modules); (C) The correlation heatmap for each module and different disease symptoms including disease severity (mild, moderate and severe) and stage (acute and recovery). The numbers in the grids represent correlation coefficients, and the numbers in parentheses represent the significance *p*-values.

the co-expression networks for RNA set1 (Fig. 5A) and RNA set2 (Fig. 5B) separately were constructed. In the co-expression network for RNA set1, there were 813 nodes (including 777 mRNAs and 36 lncRNAs) and 3773 edges. In the co-expression network for RNA set2, there were 529 nodes (including 521 mRNAs and eight lncRNAs) and 967 edges. Moreover, 10 (such as hematopoietic cell lineage and primary immunodeficiency) and 11 (such as glutathione metabolism and cytokine-cytokine receptor interaction) KEGG pathways separately were enriched for the mRNAs in the co-expression networks for RNA set1 and RNA set2 (Table 3).

**Table 1** The number and type of RNAs involved in each module.

| Module color | Number of RNAs | Number of lncRNAs | Number of mRNAs |
|---|---|---|---|
| black | 124 | 11 | 113 |
| blue | 217 | 5 | 212 |
| brown | 215 | 1 | 214 |
| green | 206 | 3 | 203 |
| greenyellow | 88 | 6 | 82 |
| grey | 379 | 17 | 362 |
| magenta | 110 | 3 | 107 |
| pink | 122 | 1 | 121 |
| purple | 91 | 2 | 89 |
| red | 182 | 15 | 167 |
| salmon | 71 | 6 | 65 |
| tan | 80 | 6 | 74 |
| turquoise | 394 | 8 | 386 |
| yellow | 210 | 19 | 191 |

**Notes.**

lncRNA, long non-coding RNA.

## Network analysis for the RSV-correlated pathways

Based on Comparative Toxicogenomics Database, 44 KEGG pathways directly related to RSV were obtained. After comparing the searched pathways and the pathways enriched for the mRNAs in the co-expression networks, four (including hsa04660:T cell receptor signaling pathway, involving CD40 ligand (*CD40LG*) and RAS guanyl releasing protein 1 (*RASGRP1*); hsa04514:Cell adhesion molecules (CAMs); hsa04650:Natural killer cell mediated cytotoxicity; and hsa04350:TGF-beta signaling pathway) and two (including hsa04060:Cytokine-cytokine receptor interaction, involving interleukin 1 beta (*IL1B*), interleukin 1 receptor type II (*IL1R2*), interleukin 18 (*IL18*) and interleukin 18 receptor 1 (*IL18R1*); and hsa04621:NOD-like receptor signaling pathway) overlapped pathways separately were obtained for the mRNAs in the co-expression networks for RNA set1 and RNA set2.

Furthermore, the networks of RSV-correlated pathways were built (Fig. 6). In the pathway network for RNA set1, long intergenic non-protein coding RNA 891 (*LINC00891*), long intergenic non-protein coding RNA 526 (*LINC00526*) and long intergenic non-protein coding RNA 1215 (LINC01215) were co-expressed with *CD40LG*, and *LINC01215* was co-expressed with *RASGRP1*. Moreover, long intergenic non-protein coding RNA 1503 (*LINC01503*) and *SMIM25* co-expressed with *IL1B*, colorectal neoplasia differentially expressed (*CRNDE*) co-expressed with *IL1R2*, BAIAP2 antisense RNA 1 (*BAIAP2-AS1*) and *CRNDE* co-expressed with *IL18*, and *SMIM2 5* co-expressed with *IL18R1* were involved in the pathway network for RNA set2.

**Table 2  The Gene Ontology (GO) terms (A) and pathways (B) separately enriched for the mRNAs in RNA set1 and RNA set2.**

**(A)**

| Category | Term | Count | *P*-value | FDR |
|---|---|---|---|---|
| RNA set1 | GO:0006350~transcription | 155 | 8.10E-12 | 1.81E-08 |
| | GO:0046649~lymphocyte activation | 34 | 2.86E-11 | 3.20E-08 |
| | GO:0045449~regulation of transcription | 178 | 7.42E-11 | 5.54E-08 |
| | GO:0045321~leukocyte activation | 34 | 5.22E-09 | 2.92E-06 |
| | GO:0030098~lymphocyte differentiation | 21 | 1.52E-08 | 6.81E-06 |
| | GO:0042110~T cell activation | 23 | 2.25E-08 | 8.41E-06 |
| | GO:0001775~cell activation | 35 | 1.07E-07 | 3.42E-05 |
| | GO:0030217~T cell differentiation | 15 | 5.89E-07 | 1.65E-04 |
| | GO:0002521~leukocyte differentiation | 21 | 9.30E-07 | 2.31E-04 |
| | GO:0030097~hemopoiesis | 27 | 1.27E-05 | 2.84E-03 |
| | GO:0006355~regulation of transcription, DNA-dependent | 113 | 2.25E-05 | 4.57E-03 |
| | GO:0048534~hemopoietic or lymphoid organ development | 28 | 2.54E-05 | 4.73E-03 |
| | GO:0002520~immune system development | 29 | 2.77E-05 | 4.76E-03 |
| | GO:0006955~immune response | 54 | 3.95E-05 | 5.89E-03 |
| | GO:0051252~regulation of RNA metabolic process | 114 | 3.70E-05 | 5.91E-03 |
| | GO:0042113~B cell activation | 13 | 1.00E-04 | 1.39E-02 |
| | GO:0002684~positive regulation of immune system process | 24 | 2.89E-04 | 3.73E-02 |
| RNA set2 | GO:0006952~defense response | 68 | 5.40E-17 | 1.32E-13 |
| | GO:0009611~response to wounding | 55 | 1.48E-12 | 1.81E-09 |
| | GO:0009617~response to bacterium | 29 | 1.73E-10 | 1.40E-07 |
| | GO:0006954~inflammatory response | 36 | 3.71E-09 | 2.26E-06 |
| | GO:0006955~immune response | 57 | 3.78E-09 | 1.84E-06 |
| | GO:0001817~regulation of cytokine production | 24 | 9.50E-08 | 3.86E-05 |
| | GO:0042742~defense response to bacterium | 18 | 3.59E-07 | 1.25E-04 |
| | GO:0006935~chemotaxis | 21 | 8.54E-07 | 2.60E-04 |
| | GO:0042330~taxis | 21 | 8.54E-07 | 2.60E-04 |
| | GO:0007599~hemostasis | 16 | 5.45E-06 | 1.47E-03 |
| | GO:0002237~response to molecule of bacterial origin | 14 | 8.97E-06 | 2.18E-03 |
| | GO:0050817~coagulation | 15 | 1.28E-05 | 2.84E-03 |
| | GO:0007596~blood coagulation | 15 | 1.28E-05 | 2.84E-03 |
| | GO:0042060~wound healing | 21 | 1.33E-05 | 2.69E-03 |
| | GO:0006690~icosanoid metabolic process | 10 | 3.17E-05 | 5.93E-03 |
| | GO:0050878~regulation of body fluid levels | 17 | 3.61E-05 | 6.26E-03 |
| | GO:0033559~unsaturated fatty acid metabolic process | 10 | 6.25E-05 | 1.01E-02 |
| | GO:0032496~response to lipopolysaccharide | 12 | 7.47E-05 | 1.13E-02 |
| | GO:0045087~innate immune response | 16 | 1.03E-04 | 1.47E-02 |
| | GO:0007626~locomotory behavior | 23 | 2.79E-04 | 3.51E-02 |

**(B)**

| Category | Term | Count | *P*-value |
|---|---|---|---|
| RNA set1 | hsa05340:Primary immunodeficiency | 11 | 6.61E-07 |

**Table 2** (*continued*)

|  | (B) |  |  |
| Category | Term | Count | *P*-value |
| --- | --- | --- | --- |
|  | hsa04662:B cell receptor signaling pathway | 10 | 2.84E-03 |
|  | hsa04514:Cell adhesion molecules (CAMs) | 13 | 6.34E-03 |
|  | hsa05330:Allograft rejection | 6 | 1.35E-02 |
|  | hsa03010:Ribosome | 9 | 2.26E-02 |
|  | hsa04650:Natural killer cell mediated cytotoxicity | 11 | 4.01E-02 |
|  | hsa04612:Antigen processing and presentation | 8 | 4.78E-02 |
| RNA set2 | hsa04610:Complement and coagulation cascades | 9 | 4.69E-03 |
|  | hsa04142:Lysosome | 12 | 5.12E-03 |
|  | hsa04060:Cytokine-cytokine receptor interaction | 20 | 5.45E-03 |
|  | hsa04664:Fc epsilon RI signaling pathway | 9 | 9.80E-03 |
|  | hsa04620:Toll-like receptor signaling pathway | 10 | 1.53E-02 |
|  | hsa04666:Fc gamma R-mediated phagocytosis | 9 | 2.92E-02 |
|  | hsa04621:NOD-like receptor signaling pathway | 7 | 3.11E-02 |
|  | hsa04722:Neurotrophin signaling pathway | 10 | 4.89E-02 |

**Notes.**
FDR,  false discovery rate.

# DISCUSSION

In this study, 2,489 DE-RNAs between the two groups of samples were obtained, including 2386 DE-mRNAs (1393 up-regulated and 993 down-regulated) and 103 DE-lncRNAs (62 up-regulated and 41 down-regulated). The black, salmon, blue, tan and turquoise modules were significantly correlated with stage, and the RNAs in them were taken as RNA set1. Meanwhile, brown, blue, magenta and pink modules were significantly related to disease severity, and the RNAs in these modules were considered as RNA set2. In the pathway network for RNA set1, *LINC00891*, *LINC00526* and *LINC01215* were co-expressed with *CD40LG*, and *LINC01215* was co-expressed with *RASGRP1*. In the pathway network for RNA set2, *LINC01503* and *SMIM25* co-expressed with *IL1B*, *CRNDE* co-expressed with *IL1R2*, *BAIAP2-AS1* and *CRNDE* co-expressed with *IL18*, and *SMIM25* co-expressed with *IL18R1* were found.

After RSV infection, a large number of cytokines, chemokines, reactive oxygen species and other active mediators are induced in cells (*Hansdottir et al., 2010*). Natural immunity and regulatory immune responses are activated by identifying the conserved structures of corresponding antigens in bacteria, viruses and the environment (*Shayakhmetov, 2010*; *Sigurs et al., 2000*). Immune response in the body changes dynamically all the time during viral infection (*Thompson et al., 2011*). This study analyzed the blood samples from infants in acute phase or recovery period of RSV infection, which was the clearance process after virus infection. After removing the virus *in vivo*, apoptosis of the effector T cells occurred and immune response of the T cells decreased (*Jie et al., 2011*).

In the pathway network for RNA set1, *CD40LG* and *RASGRP1* were involved in the T cell receptor signaling pathway in our present study. *CD40LG* promotes virus clearance ability of DNA vaccines encoding the F glycoprotein and enhances the immune response to RSV infection, therefore, *CD40LG* can strengthen the durability of DNA vaccines against RSV

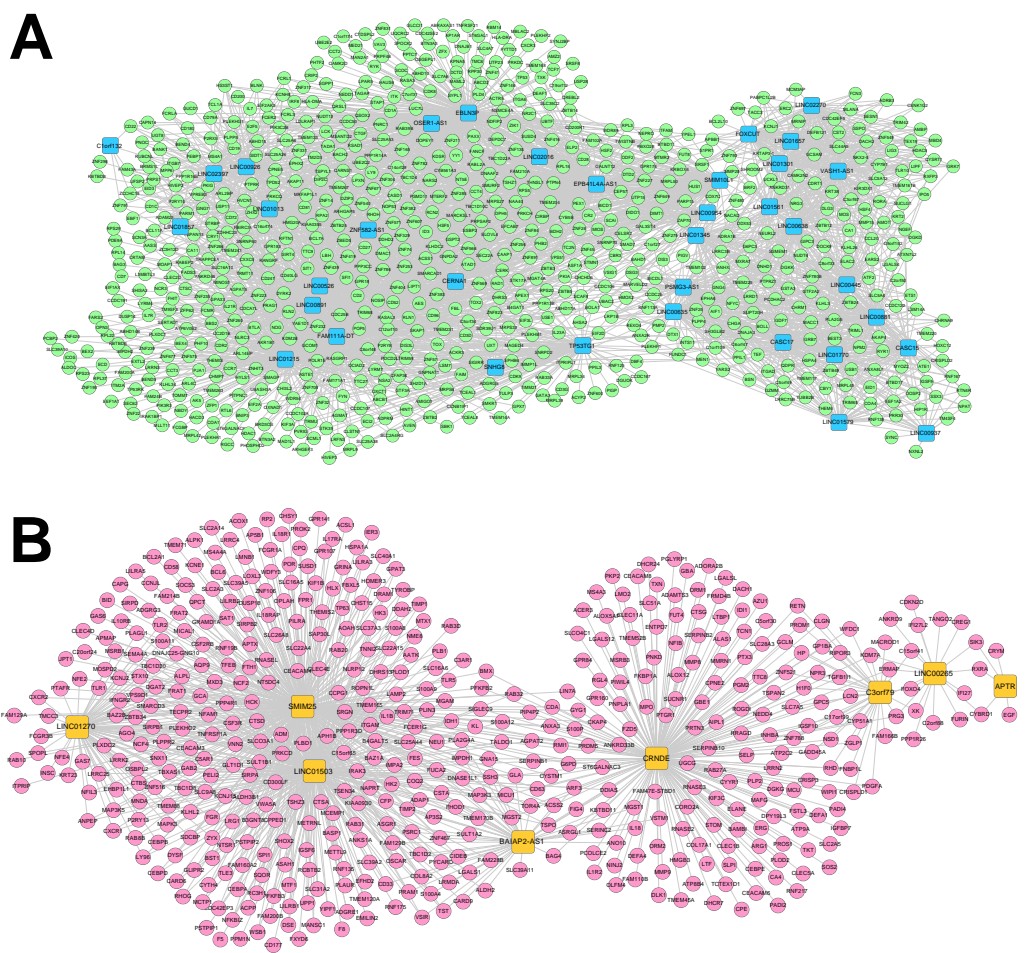

**Figure 5  Co-expression networks.** (A) The co-expression network for RNA set1. (B) The co-expression network for RNA set2. Blue and green represent RNAs that are significantly down-regulated in the samples in the acute phase of infection. Red and orange represent RNAs that are significantly up-regulated in the samples in the acute phase of infection. Squares and circles separately represent long non-coding RNAs (lncRNAs) and mRNAs.

infection (*Harcourt et al., 2003*). *RASGRP1* deficiency is related to reduced extracellular signal-regulated kinase (*ERK*) phosphorylation in B cells and T cells, and can lead to defective proliferation, motility, and activation of the cells (*Salzer et al., 2016*). Therefore, the fact that *LINC00891*, *LINC00526* and *LINC01215* co-expressed with *CD40LG*, as well as *LINC01215* co-expressed with *RASGRP1* in our study might suggest the potential function of them in the rehabilitation mechanisms of RSV infection via the T cell receptor signaling pathway.

In the pathway network for RNA set2, *IL1B*, *IL1R2*, *IL18*, and *IL18R1* were implicated in cytokine-cytokine receptor interaction in our study. Proinflammatory cytokines are found to be associated with the progression of cerebral white matter injury (WMI) in preterm infants, and cytokine-receptor interaction may be critical in determining the effects of inflammation in the development of the disease (*Bass et al., 2008*). *IL18* is a cytokine that

**Table 3** The pathways enriched for the mRNAs in the co-expression networks for RNA set1 and RNA set2.

| Category | Term | Count | *P*-value |
|---|---|---|---|
| RNA set1 | hsa04640:Hematopoietic cell lineage | 15 | 2.25E-06 |
| | hsa05340:Primary immunodeficiency | 9 | 2.89E-05 |
| | hsa04660:T cell receptor signaling pathway | 15 | 3.41E-05 |
| | hsa04662:B cell receptor signaling pathway | 10 | 1.60E-03 |
| | hsa03010:Ribosome | 9 | 1.41E-02 |
| | hsa04514:Cell adhesion molecules (CAMs) | 9 | 1.12E-02 |
| | hsa04650:Natural killer cell mediated cytotoxicity | 9 | 1.15E-02 |
| | hsa05330:Allograft rejection | 4 | 1.45E-02 |
| | hsa00600:Sphingolipid metabolism | 4 | 1.72E-02 |
| | hsa04350:TGF-beta signaling pathway | 6 | 2.15E-02 |
| RNA set2 | hsa00480:Glutathione metabolism | 7 | 3.50E-03 |
| | hsa04060:Cytokine-cytokine receptor interaction | 17 | 4.80E-03 |
| | hsa00600:Sphingolipid metabolism | 6 | 5.70E-03 |
| | hsa00052:Galactose metabolism | 5 | 6.90E-03 |
| | hsa04142:Lysosome | 10 | 7.90E-03 |
| | hsa04621:NOD-like receptor signaling pathway | 7 | 1.00E-02 |
| | hsa04640:Hematopoietic cell lineage | 8 | 1.39E-02 |
| | hsa00010:Glycolysis/Gluconeogenesis | 6 | 3.26E-02 |
| | hsa00500:Starch and sucrose metabolism | 5 | 3.56E-02 |
| | hsa00030:Pentose phosphate pathway | 4 | 3.72E-02 |
| | hsa00561:Glycerolipid metabolism | 5 | 4.43E-01 |

can enhance antiviral immunity and decrease viral load, while the RSV/*IL18* recombinant deteriorates pulmonary viral infection (*Harker et al., 2010*). RSV-induced bronchiolitis is correlated with the occurrence of atopic and allergy asthma, and *IL12* and *IL18* play critical roles in Th1 and/or Th2 immune responses to airway inflammation induced by RSV infection (*Wang et al., 2004*). *IL1α* can boost *IL8* expression in RSV-infected epithelial cells, and *IL1α* inhibition may be applied for repressing the inflammation correlated with *IL1α* and *IL8* (*Patel et al., 2010*). *IL1α* is the main cytokine secreted by RSV-infected epithelial cells, and the endothelial cell activation mediated by *IL1α* may contribute to the initiation of RSV-associated leukocyte inflammation (*Chang, Huang & Anderson, 2003*). Thus, the cytokines in *IL1* and *IL18* families might be correlated with clearance process of RSV infection through cytokine-cytokine receptor interaction, and *BAIAP2-AS1*, *CRNDE*, *LINC01503* and *SMIM25* co-expressed with the cytokines might be also involved in the rehabilitation process of RSV infection.

## CONCLUSIONS

In conclusion, a total of 2,489 DE-RNAs between the acute and recovery phase of RSV infection were screened. We identified that *LINC00891*, *LINC00526* and *LINC01215* co-expressed with *CD40LG*, as well as *LINC01215* co-expressed with *RASGRP1* might affect the rehabilitation process of RSV infection through the T cell receptor signaling

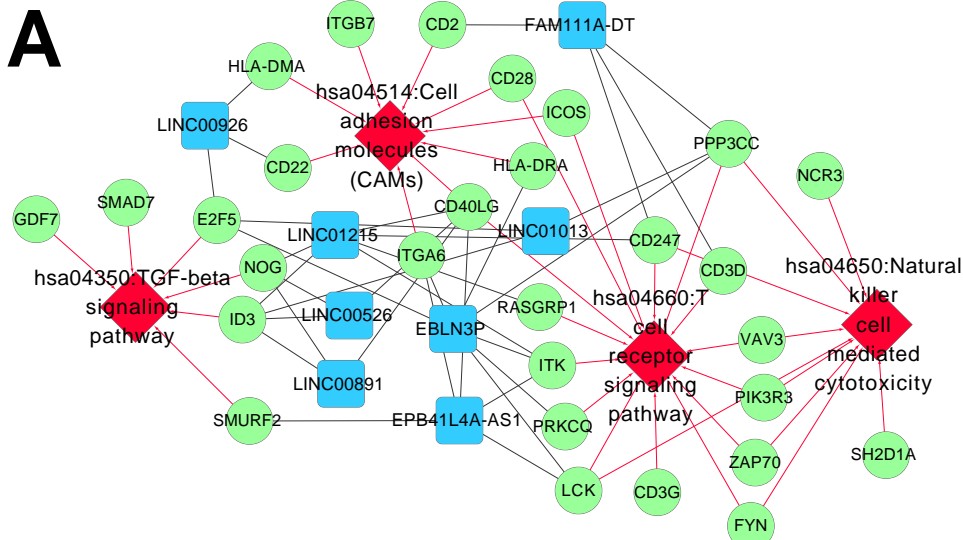

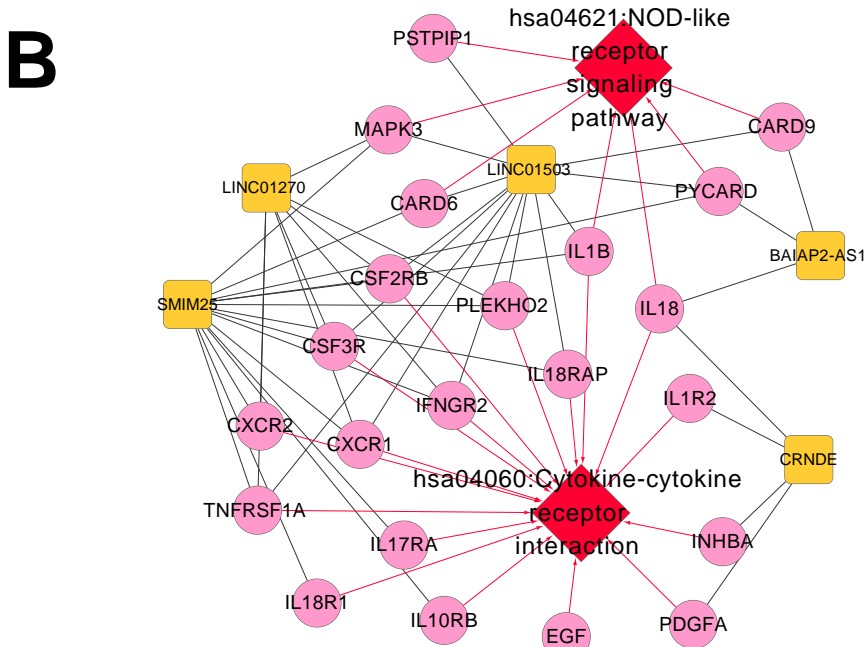

**Figure 6 Networks of respiratory syncytial virus (RSV)-correlated pathways.** (A) The network of RSV-correlated pathways for RNA set1. (B) The network of RSV-correlated pathways for RNA set2. Blue and green represent RNAs that are significantly down-regulated in the samples in the acute phase of infection. Red and orange represent RNAs that are significantly up-regulated in the samples in the acute phase of infection. Squares and circles separately represent long non-coding RNAs (lncRNAs) and mRNAs. Red diamonds represent the overlapped pathways.

pathway. Furthermore, *BAIAP2-AS1*, *CRNDE*, *LINC01503* and *SMIM25* co-expressed with the cytokines in *IL1* and *IL18* families might act in the rehabilitation mechanisms of RSV infection via cytokine-cytokine receptor interaction. However, these findings obtained from bioinformatics analyses should be further validated by experimental researches.

### Funding
This work was supported by Correlation between vitamin D level in mother/newborn and neonatal respiratory distress syndrome of Jiangsu (No.TKKY0716). The funders had no role in study design, data collection and analysis, decision to publish, or preparation of the manuscript.

### Grant Disclosures
The following grant information was disclosed by the authors:
Correlation between vitamin D level in mother/newborn and neonatal respiratory distress syndrome of Jiangsu: TKKY0716.

### Competing Interests
The authors declare there are no competing interests.

### Author Contributions
- Zuanhao Qian conceived and designed the experiments, authored or reviewed drafts of the paper, approved the final draft.
- Zhenglei Zhang and Yingying Wang performed the experiments, analyzed the data, contributed reagents/materials/analysis tools, prepared figures and/or tables, authored or reviewed drafts of the paper, approved the final draft.

### Data Availability
Raw data is available at EBI Array Express database under accession number E-MTAB-5195 here: https://www.ebi.ac.uk/arrayexpress/experiments/E-MTAB-5195/.

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
