# Peer review of "T cell receptor signaling pathway and cytokine-cytokine receptor interaction affect the rehabilitation process after respiratory syncytial virus infection"

_PeerJ, doi:10.7717/peerj.7089_

## Round 0.1 · original submission · Major Revisions

Thank you for submitting your article and I apologize for the length of time the reviews have taken. Some concerns have been raised by the reviewers, especially reviewer #2 and I ask that you address the concerns.

Reviewer 1 ·

Basic reporting

Pass. The manuscript is in clear and unambiguous English. Need improved proofreading due to some grammarical errors.

Experimental design

The acute and relapsing phase data seem from different patients. Is there a way to include same patient acute vs. relapsing dataset?

Pass

Validity of the findings

No comment

Reviewer 2 ·

Basic reporting

• Qian et al. have pulled an RSV patient sample dataset from Jong et al. 2016 (E-MTAB-5195) to compare gene changes between 39 acute phase samples vs. 21 recovery timepoint samples. In this reinterpretation of the Jong et al. dataset, the authors have identified genes that associate with the sample time-point (RNA set 1) and staging (RNA set 2). In RSV infection, a subset of individuals will be hospitalized due to severe infection complications and the Jong et al. 2016 dataset is taken specifically from individuals who are hospitalized. The authors should clarify these points and add additional information to the introduction to discuss hospitalization and severe RSV infection.
• The authors provide an immunological background of long-noncoding RNAs (lncRNAs) and immune mediators in the context of RSV. The authors should more clearly define what is known between acute and recovery stages of the in the introduction. This information will help shape their overall goal of assessing acute vs recovered patient samples.

Experimental design

• The authors have clearly stated the forms of analysis and corresponding programs that were utilized for the analysis of the Jong et al. 2016 dataset. However, the authors do not discuss how their analysis differs from the Jong et al. 2016 report. For instance, why have the authors chosen a cutoff of “|log fold change (FC)| < 0.263” compared to the Jong et al. 2016 cutoff of “absolute fold change (FC) threshold of 2”? The authors should clearly define the reasoning for their choices and compare these to the primary Jong et al. 2016 analysis to differentiate their analyses.
• “RNA set 1” and “RNA set 2”, and the relationship to the color modules, are not clearly defined early in the paper. The color modules are not clearly defined in the paper. Please provide additional information on how these groups are generated, different, and defined earlier in the paper.

Validity of the findings

• Qian et al. have performed differential gene analysis on the Jong et al. 2016 dataset to define differences between acute infected samples and recovery samples. As part of this analysis, Qian at al. have identified a unique set of genes that are different between acute vs recovery that may help inform rehabilitation mechanisms during RSV infection. However, there are key deficiencies in the analysis and discussion of the reinterpretation of the Jong el al. 2016 dataset. The Jong et al. 2016 dataset split hospitalized RSV patients into several categories that affected clustering including: age, sex, mild RSV, moderate RSV, and severe RSV. The authors have not commented how their data analysis took these factors into account. Jong et al. demonstrated that healthy controls and recovered samples are similar (1 differential expressed gene), and it is unclear how the authors analysis of acute vs recovered samples differs from the Jong et al. analysis to healthy controls. Please provide additional information how the analysis and findings of the study herein compare to the Jong et al. 2016 study.

Additional comments

• Qian et al. have provided an extended analysis of acute RSV infected samples vs recovery samples from the Jong et al. 2016 study (E-MTAB-5195). The differentially regulated genes identified in this study are discussed in the context of immune responses that might foster the rehabilitation process. However, it is not clear how the results herein are different from the Jong et al. analysis of the same data set. Further, the authors do not discuss why their analysis yielded a different set of differentially expressed genes than the Jong et al. study.
• Taken together, at this time this manuscript is not suitable for publication.

Reviewer 3 ·

Basic reporting

In this manuscript the authors utilized a pre-existing dataset obtained from a previously published study to examine the blood samples from infants undergoing an acute RSV infection as well as samples obtained following resolution of the infection. A total of 2489 differentially expressed RNAs were identified between the acute and recovery phase of the infection and were further interrogated using pathway analyses. Several RNAs associated were T cell receptor signaling were identified that may be involved in the recovery phase as well as RNAs involved in the IL-1 and IL-18 cytokine and cytokine signaling pathways.

Experimental design

The experimental design is straightforward analysis of a pre-existing dataset generated from a previous study.

Validity of the findings

No comment.

---

## Round 0.2 · accepted · Accept

Thank you for your submission. Your responses to the reviewer comments have been accepted.